# Vision State Space Duality for Medical Image Segmentation: Enhancing Precision through Non-Causal Modeling

## Abstract

In medical image analysis, Convolutional Neural Networks (CNNs) and Vision Transformers (ViTs) have set significant benchmarks. However, CNNs exhibit limitations in long-range modeling capabilities, whereas Transformers are hampered by their quadratic computational complexity. Recently, State Space Models (SSMs) have gained prominence in vision tasks as they offer linear computational complexity. State Space Duality (SSD), an improved variant of SSMs, was introduced in Mamba2 to enhance model performance and efficiency. Inspired by this, we have tailored the Vision State Space Duality (VSSD) model for medical image segmentation tasks by integrating it within a UNet-like architecture, which is renowned for its effectiveness in the field. Our modified model, named VSSD-UNet, employs skip connections to preserve spatial information and utilizes a series of VSSD blocks for feature extraction. In addition, VSSD-UNet employs a hybrid structure of VSSD and self-attention in the decoder part, ensuring that both local details and global contexts are captured. Finally, we conducted comparative and ablation experiments on two public lesion segmentation datasets: ISIC2017 and ISIC2018. The results show that VSSD-UNet outperforms several types of UNet in medical image segmentation under the same hyper-parameter setting. Our code will be released soon.

## 1 Introduction

In the medical imaging domain, segmentation is vital for advancing clinical diagnostics, informing treatment strategies, and enabling a deeper understanding of anatomical and pathological characteristics. The ability to accurately segment images into distinct regions corresponding to different tissues, organs, or abnormalities is crucial for a range of medical applications, from oncology to neurology. The integration of deep learning techniques, particularly Convolutional Neural Networks (CNNs) LeCun & Bengio (1998), has marked a significant leap forward in the accuracy and efficiency of medical image segmentation. CNNs have demonstrated their prowess in capturing local features and spatial hierarchies, leading to significant improvements in segmentation tasks. Furthermore, the advent of Vision Transformers (ViTs) Dosovitskiy et al. (2020) has introduced a new paradigm, harnessing self-attention mechanisms to capture global dependencies and long-range interactions within images, which is particularly beneficial for understanding the complex patterns present in medical imaging data.

Despite the remarkable achievements of CNNs and ViTs, there are inherent challenges that limit their effectiveness in medical image segmentation. CNNs, while excellent at capturing local features, often struggle to model long-range spatial dependencies that are essential for accurately segmenting large or complex anatomical structures. This limitation can result in segmentation inaccuracies. On the other hand, ViTs, despite their ability to provide a more comprehensive view of the image, are hindered by their quadratic computational complexity. This complexity becomes a significant bottleneck when scaling to the high-resolution images that are common in medical imaging, where detailed and precise segmentation is critical for clinical decision-making. The computational demands of ViTs can be prohibitive, particularly in time-sensitive clinical settings where real-time processing is desirable. Consequently, how to efficiently enhance the long-range dependency remains an open question.

Recently, structured state-space models (SSMs) Gu (2023); Gu et al. (2021b) inspired by classical state-space models have garnered significant interest for their computational efficiency and excellent performance in modeling long-range dependencies. Notably, Mamba, a state-of-the-art selective structured state-space model, addresses the inherent limitations of previous SSMs. It successfully demonstrates efficiency and effectiveness in long sequence modeling and achieves cutting-edge performance in continuous long sequence data analysis, such as in natural language processing and genomic analysis. Internally, Mamba integrates time-varying parameters and employs a novel hardware-aware algorithm for highly efficient training and inference, thereby avoiding the high quadratic computational complexity caused by self-attention mechanisms. Recent studies have tentatively delved into the effectiveness of SSMs across a range of visual tasks, including ImageNet classification Zhu et al. (2024b); Liu et al. (2024b), classifying remote sensing images Chen et al. (2024a), image dehazing Zheng & Wu (2024), analyzing point clouds Liang et al. (2024), and segmenting medical images Ruan & Xiang (2024b); Ma et al. (2024). This inspires us to explore the potential of using Mamba blocks to enhance long-range dependency modeling in medical image segmentation tasks.

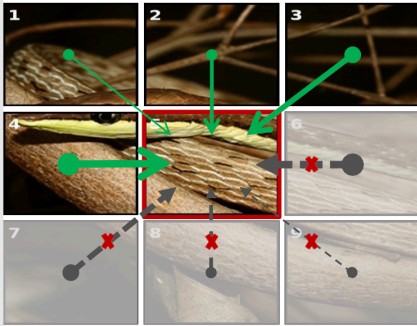

Figure 1: Two challenges when applying SSM/SSD to image data.

However, there exists a major concern regarding the application of SSD/SSMs in vision tasks, where the image data is naturally non-causal while SSD/SSMs have inherent causal properties. While another concern is flattening 2D feature maps into 1D sequences disrupts the inherent structural relationships among patches. We provide an illustration in Fig. 1 to facilitate a more intuitive understanding of these two concerns. In this example, the central token within the flattened 1D sequences is restricted to accessing only previous tokens, unable to integrate information from subsequent tokens. Additionally, the token 1, which is adjacent to the central token in the 2D space, becomes distantly positioned in the 1D sequence, disrupting the natural structural relationships.

In this work, we introduce VSSD-UNet, a model that integrates Vision State Space Duality (VSSD) within a UNet-like architecture, a framework known for its effectiveness in medical image segmentation. VSSD-UNet leverages the non-causal properties of VSSD to capture both local and global features within medical images effectively. It employs skip connections to preserve spatial hierarchies and integrates VSSD blocks for feature extraction, ensuring that the model can extract fine details while maintaining a broader contextual understanding. In addition, we employ a hybrid structure of VSSD and self-attention in the decoder part. Building on these techniques, our model provides superior segmentation performance while maintaining computational efficiency, addressing the limitations of existing models. The VSSD-UNet model represents a significant advancement in the field of medical image segmentation, offering a potential solution to the challenges faced by current deep learning models.

In summary, this paper presents several key contributions to the field of medical image segmentation. Firstly, we introduce VSSD-UNet, a novel model tailored for medical image segmentation that combines the strengths of VSSD and UNet architectures. Secondly, we provide a comprehensive evaluation of VSSD-UNet against existing segmentation models on standardized medical imaging datasets, demonstrating its superior performance. Finally, we offer an in-depth analysis of the model's computational efficiency and accuracy, highlighting its potential for real-world clinical applications.

## 2 RELATED WORKS

### 2.1 MEDICAL IMAGE SEGMENTATION

Medical image segmentation is a critical task that entails the pixel-wise classification of various anatomical structures, such as lesions, tumors, or organs, across diverse imaging modalities like endoscopy, MRI, or CT scans Chen et al. (2021). U-shaped networks Ronneberger et al. (2015); Oktay et al. (2018); Zhou et al. (2018); Huang et al. (2020); Lou et al. (2021); Ibtehaz & Kihara (2023); Chen et al. (2022); et al. (2021) have become particularly popular due to their straightforward yet effective encoder-decoder architecture. The UNet Ronneberger et al. (2015), a seminal work in this area, employs skip connections to effectively fuse features at different resolution levels. This design has been further refined by UNet++ Zhou et al. (2018), which introduces nested encoder-decoder pathways with dense skip connections, and UNet 3+ Huang et al. (2020), which presents comprehensive skip pathways for full-scale feature integration. DC-UNet Lou et al. (2021) pushes the envelope by integrating a multi-resolution convolution scheme and residual paths into its skip connections. The DeepLab series, including DeepLabv3 Chen et al. (2017) and DeepLabv3+ Chen et al. (2018), leverages atrous convolutions and spatial pyramid pooling to effectively handle multi-scale information. SegNet Badrinarayanan et al. (2017) utilizes pooling indices for feature map upsampling, ensuring boundary detail preservation. The nnU-Net et al. (2021) automatically tailors hyperparameters based on dataset-specific characteristics and employs standard 2D and 3D UNets. These U-shaped models have collectively set a high benchmark in the field of medical image segmentation.

In recent years, vision transformers have emerged as a powerful force in medical image segmentation, capable of capturing pixel relationships at a global scale Cao et al. (2021); Chen et al. (2021); Dong et al. (2021); Rahman & Marculescu (2023a;b); Wang et al. (2022a); Zhang et al. (2021); Xie et al. (2021). TransUNet Chen et al. (2021) represents a novel fusion of CNNs for local feature extraction and transformers for global context understanding, thereby enhancing the capture of both local and global features. Swin-Unet Cao et al. (2021) further extends this concept by integrating Swin Transformer blocks Liu et al. (2021) into a U-shaped model for both encoding and decoding processes. Drawing on these ideas, MERIT Rahman & Marculescu (2023b) introduces a multi-scale hierarchical transformer that employs self-attention across various window sizes, thereby enhancing the model's ability to capture multi-scale features that are crucial for medical image segmentation. These advances demonstrate the potential of transformers to significantly impact the field of medical image analysis.

### 2.2 VISION TRANSFORMERS

The emergence of Vision Transformers (ViTs) Dosovitskiy et al. (2020); Liu et al. (2021); Wang et al. (2021); Dong et al. (2022); Touvron et al. (2021) has reinvigorated the computer vision domain, a field that was once predominantly governed by Convolutional Neural Networks (CNNs) Krizhevsky et al. (2012); Simonyan & Zisserman (2014); He et al. (2016); Xie et al. (2017); Huang et al. (2019); Howard et al. (2017); Tan & Le (2019); Liu et al. (2022b). However, the self-attention mechanism in ViTs, which entails quadratic computational complexity, presents considerable difficulties when dealing with high-resolution imagery, necessitating substantial computational resources. To surmount this challenge, various strategies have been introduced, such as hierarchical model structures Liu et al. (2021; 2022a); Dong et al. (2022); Wang et al. (2021; 2022c); Han et al. (2021), windowed attention techniques Liu et al. (2021); Hassani et al. (2023); Tu et al. (2022); Zhu et al. (2023), and alternative forms of self-attention mechanisms Wang et al. (2022b); Xia et al. (2023); Yu et al. (2022). Additionally, linear attention methods Katharopoulos et al. (2020); Choromanski et al. (2020); Qin et al. (2022); Han et al. (2024) have managed to scale down the computational complexity to a linear rate by reordering the self-attention's query, key, and value operations. Yet, despite these improvements, the efficacy of linear attention still lags behind that of the quadratic self-attention Vaswani et al. (2017) and its derivatives Hassani et al. (2023); Fan et al. (2024); Zhu et al. (2023).

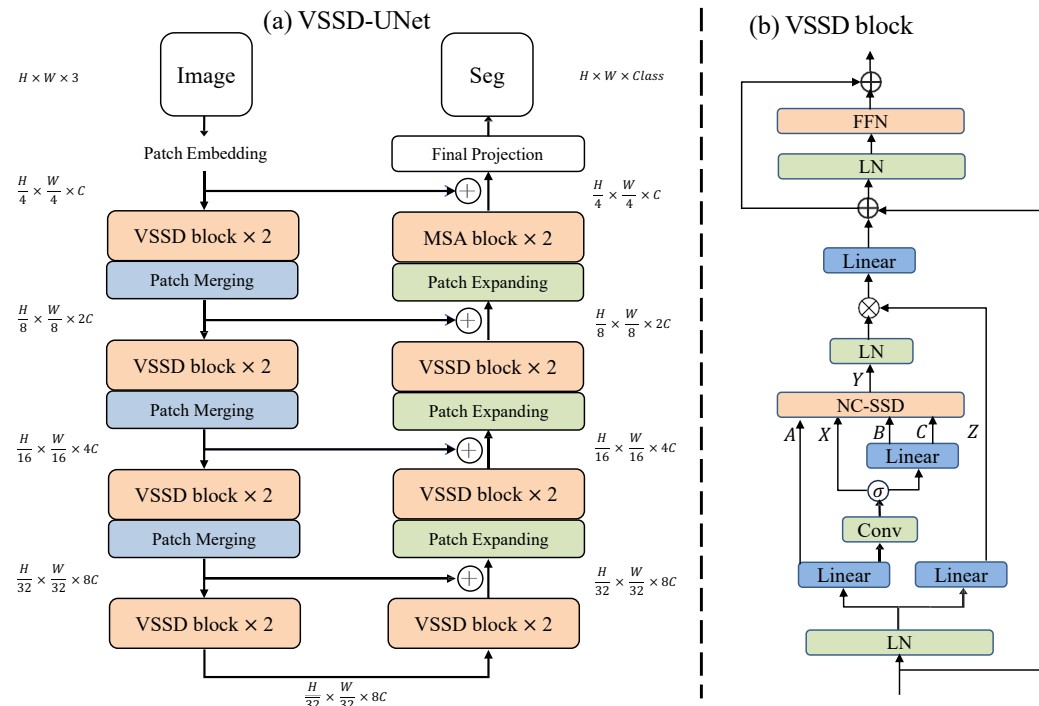

Figure 2: The architecture of VSSD-UNet, which is composed of encoder, bottleneck, decoder and skip connections. The encoder, bottleneck and decoder are all constructed based on Visual Mamba block.

## 2.3 STATE SPACE MODELS

State Space Models (SSMs) Gu et al. (2020; 2021c;a); Smith et al. (2022); Fu et al. (2022); Gu & Dao (2023) have garnered significant research interest due to their expansive receptive fields and linear computational complexity. A notable SSM, Mamba Gu & Dao (2023), has introduced the S6 block, which has demonstrated comparable or superior performance to transformers in Natural Language Processing (NLP) tasks. This has spurred further explorations Pei et al. (2024); Huang et al. (2024); Du et al. (2024); Yang et al. (2024a); Chen et al. (2024b); Li et al. (2024); Yang et al. (2024b); Ruan & Xiang (2024b) into adapting the S6 block for visual tasks, with studies showing it can compete with both CNNs and Vision Transformer (ViT) models. However, a key challenge in developing Mamba-based models for computer vision lies in aligning the model's causal nature with the non-causal aspects of image data. A common strategy to overcome this is to flatten 2D feature maps into 1D sequences using various scanning methods before processing them through the S6 block. These diverse scanning approaches have been proven effective across multiple studies Zhu et al. (2024a); Liu et al. (2024a); Huang et al. (2024); Pei et al. (2024); Shi et al. (2024). Recently, Mamba2 Dao & Gu (2024) has identified a close relationship between SSMs and structured masked attention, establishing them as dual concepts and introducing State Space Duality (SSD). We extend this work to show that SSD can be adapted into a non-causal model through a simple transformation, eliminating the need for specific scanning routes.

## 3 METHODS

### 3.1 ARCHITECTURE OVERVIEW

The architecture of the proposed VSSD-UNet is outlined in Figure 2. The input images are first divided into patches similar to ViT and VMamba and transformed into sequences. An initial linear embedding layer adjusts feature dimensions to an arbitrary size denoted as $C$. These patch tokens are processed by several VSSD blocks and patch merging layers to generate hierarchical features.

The patch merging layers are responsible for reducing the image scale and boosting the feature dimensions, whereas the VSSD blocks concentrate on learning feature representations. The encoder produces outputs with resolutions of $\frac{H}{4} \times \frac{W}{4} \times C$, $\frac{H}{8} \times \frac{W}{8} \times 2C$, $\frac{H}{16} \times \frac{W}{16} \times 4C$, and $\frac{H}{32} \times \frac{W}{32} \times 8C$, respectively. The decoder includes VSSD and patch expanding layers to restore the feature size. It replaces the NC-SSD block with self-attention module exclusively in the last stage, recovering spatial details lost during downsampling through skip connections. Both the encoder and decoder use two VSSD blocks each. The details of VSS block, patch merging of encoder, and patch expanding of decoder is discussed in the following subsections.

## 3.2 VSSD BLOCK

### 3.2.1 PRELIMINARIES OF MAMBA

The SSM is a concept derived from modern control theory's linear time-invariant system which maps the continuous stimulation $x(t) \in \mathcal{R}$ to response $y(t) \in \mathcal{R}$. This process can be formulated through the subsequent linear ordinary differential equation (ODE),

$$
\begin{aligned}
h'(t) &= \mathbf{A}h(t) + \mathbf{B}x(t) \\
y(t) &= \mathbf{C}h(t)
\end{aligned}
\tag{1}
$$

where $\mathbf{A} \in \mathcal{R}^{N \times N}$ denotes the state matrix, while $\mathbf{B} \in \mathcal{R}^{N \times 1}$ and $\mathbf{C} \in \mathcal{R}^{N \times 1}$ are the projection parameters.

Structured State Space Sequence Model (S4) and Mamba discretize this continuous system to make it more suitable for deep learning scenarios. Specifically, they introduce a timescale parameter $\mathbf{\Delta}$ and transform $\mathbf{A}$ and $\mathbf{B}$ into discrete parameters $\overline{\mathbf{A}}$ and $\overline{\mathbf{B}}$ using a fixed discretization rule. Typically, the zero-order hold (ZOH) is employed as the discretization rule and can be defined as follows:

$$
\begin{aligned}
\bar{\mathbf{A}} &= \exp(\Delta\mathbf{A}) \\
\bar{\mathbf{B}} &= (\Delta\mathbf{A})^{-1}(\exp(\Delta\mathbf{A}) - \mathbf{I}) \cdot \Delta\mathbf{B}
\end{aligned}
\tag{2}
$$

After discretization, Eq. 1 can be rewritten as,

$$
\begin{aligned}
h_k &= \bar{\mathbf{A}}h_{k-1} + \bar{\mathbf{B}}x_k \\
y_k &= \mathbf{C}h_k
\end{aligned}
\tag{3}
$$

At last, the output can be calculated in a convolution representation, as follows,

$$
\begin{aligned}
\bar{\mathbf{K}} &= (\bar{\mathbf{C}}\bar{\mathbf{B}}, \bar{\mathbf{C}}\bar{\mathbf{A}}\bar{\mathbf{B}}, \cdots, \bar{\mathbf{C}}\bar{\mathbf{A}}^{L-1}\bar{\mathbf{B}}) \\
y &= x * \bar{\mathbf{K}}
\end{aligned}
\tag{4}
$$

where $L$ is the length of the input sequence $x$, and $\bar{\mathbf{K}} \in \mathcal{R}^L$ denotes the structured convolutional kernel.

### 3.2.2 VSSD BLOCK

State Space Duality (SSD) is an enhancement over traditional State Space Models (SSMs), offering improved performance and efficiency in processing sequence data. However, SSD inherently operates in a causal manner, which limits its applicability to non-causal vision tasks where information from future steps is just as relevant as past steps. To address this, we utilize Non-Causal SSD (NC-SSD), which modifies the role of the state transition matrix $\mathbf{A}$ to enable non-causal processing.

In the traditional SSD framework, the model updates the hidden state $h(t)$ and computes the output $y(t)$ as follows:

$$
h(t) = A_t h(t-1) + \mathbf{B}_t x(t), y(t) = \mathbf{C}_t h(t).
\tag{5}
$$

where $A$ is the state transition matrix, $B$ is the input matrix. $C$ is the output matrix.

In NC-SSD, it transforms the role of $A$ from a matrix to a scalar to facilitate non-causal processing. The key equation becomes:

$$
h(t) = h(t-1) + \frac{1}{A} \cdot \mathbf{B}_t \cdot x(t)
\tag{6}
$$

This equation shows that the current state $h(t)$ is influenced by the previous state $h(t-1)$, the input matrix $B_t$, and the current input $x(t)$, with the influence weighted by $\frac{1}{A}$.

To fully achieve non-causality, NC-SSD employs bidirectional scanning, which involves processing the data in both forward and reverse sequences. The combined hidden state $H$ from bidirectional scanning is given by:

$$\mathbf{H}_i = \sum_{j=1}^{i} \frac{1}{A_j} \mathbf{Z}_j + \sum_{j=-L}^{-1} \frac{1}{A_{i+j}} \mathbf{Z}_{i+j}. \tag{7}$$

where $Z_j = B_j \cdot x(j)$ is the transformed input for the $j$-th token in the sequence.

By integrating the results from both directions, it can be ensured that each token has access to global information, not just the tokens before it in the sequence. Assuming each token's contribution can be considered independently, the hidden state $H$ can be simplified to:

$$\mathbf{H} = \sum_{j=1}^{L} \frac{1}{A_j} \mathbf{Z}_j. \tag{8}$$

This equation shows that all tokens contribute equally to the hidden state $H$, effectively removing the causal constraint and allowing the model to process information in a non-linear sequence.

To implement VSSD efficiently, we revise the tensor contraction algorithm to:

1.Expand the input $\mathbf{X}$ using $\mathbf{B}$:

$$\mathbf{Z} = \text{contract}(\text{LD}, \text{LN} \rightarrow \text{LND})(\mathbf{X}, \mathbf{B}) \tag{9}$$

2.Unroll scalar SSM recurrences to create a global hidden state $\mathbf{H}$:

$$\mathbf{H} = \text{contract}(\text{LL}, \text{LDN} \rightarrow \text{ND})(\mathbf{M}, \mathbf{Z}) \tag{10}$$

3.Contract the hidden state $\mathbf{H}$ with $\mathbf{C}$ to produce the output $\mathbf{Y}$:

$$\mathbf{Y} = \text{contract}(\text{LN}, \text{ND} \rightarrow \text{LD})(\mathbf{C}, \mathbf{H}). \tag{11}$$

These steps replace the traditional recurrent computations with parallelizable operations, significantly enhancing training and inference speeds.

In summary, VSSD allows for more flexible processing of sequence data by removing the constraints of causality, leading to improved performance and efficiency in vision tasks.

### 3.3 ENCODER

In the encoder of the VSSD-UNet, $C$-dimensional tokenized inputs pass through two sequential VSSD blocks to extract features without changing their size or dimension. The patch merging layer is utilized for downsampling in the encoder of VSSD-UNet, reduces the token count by $\frac{1}{2}$ while doubling feature dimensions by $2\times$, by segmenting inputs into quadrants by $\frac{1}{4}$, concatenating them, and then normalizing dimensions through a layernorm each time.

### 3.4 DECODER

The decoder also uses two VSSD blocks in succession to reconstruction the features. Instead of merging layers, it uses patch expansion layers to upscale deep features. This process effectively halves feature dimensions by $\frac{1}{2}$ while enhancing image resolution ($2\times$ upscaling). It works by an initial layer that doubles feature dimensions before reorganizing and reducing them for resolution enhancement.

Moreove, Mamba2 demonstrates that integrating SSD with standard Multi-head Self Attention (MSA) yields additional improvements. In a similar way, our model incorporates self-attention. However, unlike Mamba2, which uniformly intersperses self-attention throughout the network, we strategically replace the VSSD block with self-attention module exclusively in the last stage. This modification leverages the robust capabilities of self-attention in processing high-level features, as evidenced by prior works Lin et al. (2023); Ren et al. (2023); Fan et al. (2024) in vision tasks.

Table 1: Comparative experimental results on the ISIC2017 dataset. The best results are highlighted in bold fonts. " ↑ "and " ↓ " indicate that larger or smaller is better.

| Model | Year | mIoU(%)↑ | DSC(%)↑ | Acc(%)↑ | Spe(%)↑ | Sen(%)↑ |
|---|---|---|---|---|---|---|
| UNet | 2015 | 75.97 | 86.34 | 95.53 | 97.75 | 84.47 |
| R2UNet | 2018 | 73.43 | 84.68 | 95.08 | 97.86 | 81.25 |
| UNet++ | 2019 | 77.85 | 87.55 | 95.91 | 97.94 | 85.82 |
| R2AttUNet | 2021 | 75.07 | 85.76 | 95.24 | 97.17 | 85.63 |
| SwinUnet | 2022 | 67.93 | 80.90 | 93.75 | 96.69 | 79.11 |
| MISSFormer | 2022 | 75.84 | 86.26 | 95.62 | **98.34** | 82.09 |
| MALUNet | 2022 | 74.69 | 85.51 | 95.15 | 97.10 | 85.46 |
| H2Former | 2023 | 76.27 | 86.54 | 95.58 | 97.72 | 84.90 |
| EGE-UNet | 2023 | 76.50 | 86.68 | 95.65 | 97.88 | 84.55 |
| MHorunet | 2024 | 78.16 | 87.73 | 95.77 | 97.15 | 85.99 |
| VMUNet | 2024 | 77.24 | 87.16 | 95.78 | 97.82 | 85.62 |
| VMUNet v2 | 2024 | 75.25 | 85.88 | 95.34 | 97.47 | 84.71 |
| H-vmunet | 2024 | 78.18 | 87.75 | 95.82 | 97.12 | 85.72 |
| ULVM-UNet | 2024 | 78.13 | 87.72 | 95.78 | 97.59 | 83.61 |
| VSSD-UNet | - | **78.30** | **87.83** | **96.00** | 97.99 | **86.14** |

## 3.5 BOTTLENECK & SKIP CONNETIONS

In the VSSD-UNet bottleneck, we use two VSSD blocks to process the features. At each stage of the encoder and decoder, skip connections are utilized to blend features from multiple scales with the upscaled image outputs. This process merges information from both shallow and deep layers, which enhances the spatial details in the segmentation results. After that, a linear layer is applied to keep the combined features' dimensions the same as the upsampled resolution, ensuring consistency with the upscaled resolution.

## 4 EXPERIMENTS

### 4.1 DATA SETS

In this section, we conducted extensive experiments using two prominent lesion segmentation datasets that are publicly available: the International Skin Imaging Collaboration's 2017 and 2018 challenge datasets (ISIC2017 and ISIC2018), to train and evaluate the proposed model. These datasets consist of a substantial collection of dermoscopic images, with ISIC2017 containing 2,150 images and ISIC2018 containing 2,694 images, all of which are accompanied by segmentation mask labels. Following the methods employed in prior research Ruan et al. (2022; 2023), we segmented these datasets into training and test subsets at a ratio of 7:3. To elaborate, the ISIC2017 dataset was divided into a training set of 1,500 images and a test set of 650 images. Similarly, the ISIC2018 dataset was split into a training set comprising 1,886 images and a test set comprising 808 images. This approach allowed us to train and assess the performance of our proposed model across a broad spectrum of lesion segmentation tasks.

### 4.2 IMPLEMENTATION DETAILS

We implemented our VSSD-UNet using PyTorch 1.13 and trained it on an A100-PCIE-40G GPU with 24 GB of memory for 300 epochs with a batch size of 32. The input images are uniformly resized to 224 × 224. We employed data augmentation techniques such as random flipping and random rotation to prevent overfitting. We used the AdamW optimizer with an initial learning rate of $1 \times 10^{-3}$, $\beta_1$ of 0.9, $\beta_2$ of 0.999, and weight decay of $1 \times 10^{-4}$. Additionally, we applied a cosine annealing learning rate decay strategy and an early stopping mechanism. To ensure reproducibility, we set the random seed to 42.

Table 2: Comparative experimental results on the ISIC2018 dataset. The best results are highlighted in bold fonts. " ↑ "and " ↓ " indicate that larger or smaller is better.

| Model | Year | mIoU(%)↑ | DSC(%)↑ | Acc(%)↑ | Spe(%)↑ | Sen(%)↑ |
|---|---|---|---|---|---|---|
| UNet | 2015 | 77.22 | 87.15 | 93.86 | 96.56 | 85.47 |
| R2UNet | 2018 | 71.74 | 83.55 | 92.36 | 96.41 | 79.74 |
| UNet++ | 2019 | 79.14 | 88.36 | 94.40 | 96.69 | 87.28 |
| R2AttUNet | 2021 | 75.24 | 85.87 | 93.15 | 95.62 | 85.47 |
| SwinUnet | 2022 | 74.26 | 85.23 | 92.87 | 95.55 | 84.54 |
| MISSFormer | 2022 | 77.94 | 87.60 | 94.11 | 96.89 | 85.48 |
| MALUNet | 2022 | 78.09 | 87.70 | 94.07 | 96.41 | 86.80 |
| H2Former | 2023 | 77.33 | 87.21 | 93.89 | 96.57 | 85.56 |
| EGE-UNet | 2023 | 78.90 | 88.20 | 94.25 | 96.17 | 88.29 |
| MHorunet | 2024 | 79.40 | 88.52 | 94.47 | 96.70 | 87.55 |
| VMUNet | 2024 | 74.14 | 85.15 | 93.03 | 96.54 | 82.10 |
| VMUNet v2 | 2024 | 78.25 | 87.80 | 94.09 | 96.25 | 87.38 |
| H-vmunet | 2024 | 79.41 | 88.52 | 94.37 | 96.03 | 89.20 |
| ULVM-UNet | 2024 | 78.74 | 88.10 | 94.29 | 96.68 | 86.85 |
| VSSD-UNet | - | **80.65** | **89.29** | **94.73** | **97.24** | **90.18** |

## 4.3 EVALUATION METRICS

We used five metrics to assess the quality of the segmentations: Mean Intersection over Union (mIoU), Dice Similarity Score (DSC), Accuracy (Acc), Sensitivity (Sen), and Specificity (Spe). The mathematical formulations for these metrics are summarized as follows:

$$mIoU = \frac{TP}{TP + FP + FN} \tag{12}$$

$$DSC = \frac{2TP}{2TP + FP + FN} \tag{13}$$

$$Acc = \frac{TP + TN}{TP + TN + FP + FN} \tag{14}$$

$$Sen = \frac{TP}{TP + FN} \tag{15}$$

$$Spe = \frac{TN}{TN + FP} \tag{16}$$

where TP, FP, FN, TN represent true positive, false positive, false negative, and true negative.

## 4.4 COMPARISON RESULTS

To validate the effectiveness of our approach, we compared VSSD-UNet with other state-of-the-art methods. Specifically, this comparison includes UNet Ronneberger et al. (2015), R2UNet Alom et al. (2018), UNet++ Zhou et al. (2019), R2AttUNet Zuo et al. (2021), SwinUnet Aghdam et al. (2023), MISSFormer Huang et al. (2023), MALUNet Ruan et al. (2022), H2Former He et al. (2023), EGEUNet Ruan et al. (2023), MHorunet Wu et al. (2024a), VMUNet Ruan & Xiang (2024a), VMUNet v2 Zhang et al. (2024), H-vmunet Wu et al. (2024b), UltraLight-VM-UNet Wu et al. (2024c), and VSSD-NUet. 1 and 2 show the comparative results on the ISIC2017 and ISIC2018 datasets, respectively. Our proposed VSSD-UNet outperformed the other models in terms of mIoU, DSC, Acc, Spe, and Sen metrics.

## 4.5 ABLATIONS

To validate the effectiveness of the proposed modules, we conducted detailed ablation experiments on the VSSD-UNet model. Using the SSD block as the token mixer and patchified downsamplers (e.g. convolution with $4 \times 4$ kernel and stride of 4 in stem) following Swin Liu et al. (2021) and vallina VMamaba Liu et al. (2024a), we established the baseline configuration, detailed in the first

row of Tab. 3. Our ablation study was conducted on an A100-PCIE-40G GPU with a batch size of 128 using FP16 precision.

Table 3: **Ablation study of VSSD-UNet.** Our VSSD consistently outperforms vallina SSD and Bi-SSD in terms of accuracy and efficiency.

| Op. Type | Downsampler | Layers | Top-1 Acc(%) | #Params | FLOPs (G) | Thru. (imgs/sec) | Train Thru. (imgs/sec) |
|---|---|---|---|---|---|---|---|
| **SSD** | Patch | 2, 4, 8, 4 | 81.0 | 14.8 M | 2.1 | 1818 | 523 |
| **Bi-SSD** | Patch | 2, 4, 8, 4 | 81.4 | 15.2 M | 2.2 | 1741 | 399 |
| **VSSD** | Patch | 2, 4, 8, 4 | 81.6 | 14.8 M | 2.1 | 1843 | 606 |
| **Hybrid** | Patch | 2, 4, 8, 4 | 82.3 | 13.4 M | 2.1 | 1890 | 622 |

**Different SSD Mechanisms.** In our ablation study for the token mixer, we explored different scanning routes for SSD. Specifically, we introduced Bi-SSD, which splits channels and reverses one part to create backward scanning sequences. These sequences are then concatenated post-SSD block. As shown in Tab. 3, our VSSD model outperforms both the vanilla SSD and Bi-SSD by 0.6% and 0.2% in top-1 accuracy, respectively. Moreover, both training and inference throughput are enhanced, with VSSD improving training throughput by nearly 50% compared to the Bi-SSD approach.

**Hybrid Architecture and Overlapped Downsampler.** The effectiveness of incorporating standard attention in the last stage is demonstrated in the last row of Tab. 3. Specifically, replacing VSSD with standard attention in the last stage results in a 0.7% improvement in accuracy while slightly reducing the parameters.

## 5  CONCLUSION

In this paper, we introduced VSSD-UNet, which is a mamba-based UNet style network for medical image segmentation. The performance demonstrates that VSSD-UNet superior performance against classical similar network such as UNet and Swin-UNet. In the future, we aim to conduct more in-depth explorations on more medical image segmentation tasks from different modalities and targets, with comparisons to more segmentation backbones. Besides, we aim to extend VSSD-UNet to 3D medical images to further enhance the developments in medical imaging.

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
