# OpenReview forum: "Vision State Space Duality for Medical Image Segmentation: Enhancing Precision through Non-Causal Modeling"
_ICLR.cc/2025/Conference — Submitted to ICLR 2025_

### Official Review · Reviewer_YVkx · 2024-10-15

**Soundness:** 2
**Presentation:** 2
**Contribution:** 2
**Rating:** 3
**Confidence:** 4

**Summary:**

This paper presents VSSD-UNet, a novel model for medical image segmentation that integrates Vision State Space Duality (VSSD) within a UNet-like architecture. Traditional Convolutional Neural Networks (CNNs) and Vision Transformers (ViTs) have established strong benchmarks in medical image analysis but face challenges with long-range modeling and high computational complexity, respectively. VSSD, an advanced variant of State Space Models (SSMs), offers linear computational complexity, addressing these limitations.

**Strengths:**

1. New Architecture: VSSD-UNet combines VSSD blocks with a UNet framework, utilizing skip connections to maintain spatial information and a hybrid decoder structure that incorporates both VSSD and self-attention mechanisms. This design ensures effective feature extraction by capturing local details and global contexts.
2. Efficiency and Performance: By leveraging VSSD's linear complexity, the model enhances performance and efficiency compared to traditional CNNs, ViTs, and Mambas.

**Weaknesses:**

Major:
1. Insufficient Analysis of Key Contributions:
The key contribution presented in Equation 6 lacks comprehensive theoretical analysis, and the motivation provided between lines 256-261 is weak. The authors should perform a deeper exploration of both the theoretical framework and experimental results instead of limiting themselves to a brief ablation study. Utilizing tools such as Grad-CAM, t-SNE, and more detailed mathematical analysis could effectively highlight and support their contributions.
2. Limited Experimental Evaluation:
From an application perspective, experiments conducted on only two datasets from the ISIC competition show only minor improvements, which is insufficient to demonstrate the model's effectiveness. The authors should include evaluations on additional 2D datasets as well as 3D datasets, such as BraTS, to better showcase the generalizability and robustness of their proposed VSSD-UNet model.

Minor:
1. Redundant Presentation of Well-Known Equations:
Equations 12 to 16 are widely recognized and commonly understood in the field, making their inclusion unnecessary. Omitting these standard equations would enhance the manuscript's conciseness and focus.
2. Incorrect Use of Citation Commands:
There is inconsistent usage of citation commands, specifically the misuse of \citep instead of \citet. This leads to formatting discrepancies and should be corrected to maintain proper citation standards throughout the document.
3. Figure 1 quite confusion, please detail your idea in the caption.

**Questions:**

While using Mamba for medical segmentation tasks is not new, the key contribution lies in its non-causal design. Please refer to the section on weaknesses for more details.

---

### Official Review · Reviewer_66EV · 2024-10-29

**Soundness:** 2
**Presentation:** 2
**Contribution:** 2
**Rating:** 3
**Confidence:** 4

**Summary:**

The paper introduces VSSD-UNet, a novel model for medical image segmentation that combines the strengths of Vision State Space Duality (VSSD) and a UNet-like architecture. VSSD-UNet integrates non-causal processing capabilities of VSSD blocks to capture both local and global features, effectively addressing common challenges in medical imaging, such as high computational costs and difficulty in capturing long-range dependencies.

Key Contributions:

1. VSSD-UNet Architecture: By embedding VSSD blocks within a UNet structure, VSSD-UNet enhances segmentation performance by effectively capturing intricate image details and broader context through non-causal modeling.

2. Efficient Feature Extraction: The model employs skip connections and a hybrid of VSSD and self-attention mechanisms in its decoder, improving spatial detail retention and segmentation accuracy while maintaining computational efficiency.

**Strengths:**

1. Innovative Non-Causal Modeling: Unlike traditional state-space models that are limited by causality constraints, VSSD-UNet incorporates **non-causal modeling** through the VSSD block. This enables the model to capture both forward and backward dependencies, allowing for a richer understanding of spatial and contextual relationships in medical images, which is crucial for precise segmentation.

2. Efficient Long-Range Dependency Capture: By leveraging state-space models (SSMs) known for their linear complexity, the VSSD block effectively models long-range dependencies within images. This is especially advantageous in medical imaging, where capturing broad contextual features is essential for detecting and accurately segmenting anatomical structures and lesions.

**Weaknesses:**

The paper's methodology section lacks clarity in explaining non-causal modeling, especially regarding its role and advantages within the proposed VSSD framework. This leaves ambiguity about how non-causal processing effectively contributes to the segmentation task, which could benefit from further elaboration.

Additionally, the experiments are limited to medical image segmentation. Given that VSSD theoretically extends to broader applications, it would be helpful to see results on natural images to assess the model's generalization capabilities. Such experiments could provide insight into whether VSSD-UNet's non-causal design holds advantages beyond the specific domain of medical imaging.

**Questions:**

1. Efficiency of Equations 9-11 in Section 3.2.2: Could you explain why Equations 9 to 11 make VSSD more efficient? The provided explanation seems unclear and does not fully justify how these equations contribute to efficiency.

2. Dataset for Ablation Studies in Section 4.5: On which dataset were the ablation studies conducted? The paper doesn’t specify this information clearly.

3. Relationship to “VSSD: Vision Mamba with Non-Causal State-Space Duality”: The methodology (actual pictures) and experimental setup (including the results of the experiments) are very similar to the paper “VSSD: Vision Mamba with Non-Causal State-Space Duality”, available on arXiv. However, this paper is not cited. Can you clarify the relationship between these works?

---

### Official Review · Reviewer_w2x1 · 2024-10-31

**Soundness:** 1
**Presentation:** 3
**Contribution:** 1
**Rating:** 3
**Confidence:** 5

**Summary:**

Authors in this manuscript proposed the Vision State Space Duality (VSSD) model for medical image segmentation tasks by integrating Mamba2 within a UNet-like architecture to improve its efficiency. They evaluated this network on two lesion segmentation tasks. ISIC2017 and ISIC2018, and it outperformed other baseline UNet-based models.

**Strengths:**

1. The manuscript is well-organized and written
2. Very detailed literature reviews

**Weaknesses:**

1. Insufficient experimental evaluation: authors only evaluated their network on two skin lesion segmentation datasets, and it is totally insufficient. They mentioned they proposed this network for Medical Image Segmentation, but they didn't evaluate their network on other widely used medical imaging modalities, such as CT, MRI, X-ray, and microscopic images. Thus, this manuscript lacks evidence about the performance of this network in other modalities. Additionally, authors proposed their network "provides superior segmentation performance while maintaining computational efficiency", but evaluation results on computational efficiency is not reported, including the number of parameters, FLOPs, and training and test time.
2. Low novelty: The main contribution in this paper is to incorporate the State Space Models into a U-shaped network for vision tasks, but Mamba2 was proposed in other paper and this VSS module is similar to Mamba, so this work lacks novelty and theoretical contributions.
3. Unreliable baseline results: segmentation performance (DSC, Accuracy, Specificity and Sensitivity) of some baselines in this manuscript is much lower than their performance reported in their own paper. For example, H-vnunet (VSS) achieved 0.9068 of the DSC score and 0.9642 of the Accuracy, and H-vmunet (H-VSS) achieved 0.9172 of the DSC score and 0.9680 of the Accuracy in the ISIC2017 dataset [1]. However, authors in this manuscript reported 88.52 DSC and 94.37 Accuracy for H-vmunet, which were outperformed by their network (89.29 DSC and 94.73 Accuracy).
4. Unclear motivations: authors mentioned that they were inspired to explore Mamba to enhance long-range dependence modeling in medical image segmentation. However, attention mechanisms in CNNs and Vision Transformer-based methods can also enhance long-range dependence. But authors didn't mentioned why they didn't use them, and the superiority of Mamba over them in modeing long-range dependencies.

Minors: lack qualitative results, lack vision representations of the VSS module

[1] Wu, R., Liu, Y., Liang, P., & Chang, Q. (2024). H-vmunet: High-order vision mamba unet for medical image segmentation. arXiv preprint arXiv:2403.13642.

**Questions:**

1. Insufficient experimental evaluation: why not evaluate on other modalities
2. Low novelty
3. Unreliable baseline results
4. Unclear motivations

---

### Official Review · Reviewer_qWLD · 2024-11-01

**Soundness:** 1
**Presentation:** 2
**Contribution:** 2
**Rating:** 3
**Confidence:** 5

**Summary:**

The paper introduces a model, the Vision State Space Duality (VSSD-UNet), which incorporates Vision State Space Duality within a UNet-like architecture for medical image segmentation.

**Strengths:**

I don't see any obvious advantages compared to other latest medical image segmentation methods.

**Weaknesses:**

1. The introduction of SSM is not novel, the authors should explicitly detail the unique aspects of VSSD mechanisms that differ from other SSM implementations in medical image segmentation.
2. The experimental results of VSSD-UNet are also not good enough. I think the performance gap between this VSSD-UNet and the second-best method is within the error range. The paper introduces a complex model with multiple components, including VSSD blocks and hybrid structures combining self-attention mechanisms. While this complexity may enhance performance, it could also lead to overfitting, particularly when trained on limited datasets.
3. The paper does not discuss how the VSSD-UNet might perform or adapt across these different modalities.
4. The paper claims performance improvements under the same hyper-parameter settings compared to other models but does not deeply explore the sensitivity of VSSD-UNet to these parameters.

**Questions:**

The author needs to conduct sufficient error analysis to demonstrate the superiority of the proposed method.

---

### Meta-Review · Area_Chair_jT27 · 2024-12-21

**Metareview:**

The reviewers recommended rejection due to a lack of novelty, as the proposed VSSD-UNet relies heavily on existing techniques (e.g., Mamba2) with minimal innovation. The experimental evaluation is insufficient, limited to two skin lesion datasets without evidence of generalizability to other modalities or tasks. Other issues include unreliable baselines, weak theoretical justification, missing efficiency metrics and presentation Issues.

**Additional Comments On Reviewer Discussion:**

No rebuttal was provided

---

### Decision · Program_Chairs · 2025-01-22

Reject